# Synchronous Analysis of Speech Production and Lips Movement to Detect Parkinson’s Disease Using Deep Learning Methods

**DOI:** 10.3390/diagnostics15010073

**Published:** 2024-12-31

**Authors:** Cristian David Ríos-Urrego, Daniel Escobar-Grisales, Juan Rafael Orozco-Arroyave

**Affiliations:** 1GITA Lab., Faculty of Engineering, University of Antioquia, Medellín 050010, Colombia; cdavid.rios@udea.edu.co (C.D.R.-U.); daniel.esobar@udea.edu.co (D.E.-G.); 2LME Lab., University of Erlangen, 91054 Erlangen, Germany

**Keywords:** Parkinson’s disease, speech analysis, lip movement analysis, fusion methods, attention mechanisms

## Abstract

Background/Objectives: Parkinson’s disease (PD) affects more than 6 million people worldwide. Its accurate diagnosis and monitoring are key factors to reduce its economic burden. Typical approaches consider either speech signals or video recordings of the face to automatically model abnormal patterns in PD patients. Methods: This paper introduces, for the first time, a new methodology that performs the synchronous fusion of information extracted from speech recordings and their corresponding videos of lip movement, namely the bimodal approach. Results: Our results indicate that the introduced method is more accurate and suitable than unimodal approaches or classical asynchronous approaches that combine both sources of information but do not incorporate the underlying temporal information. Conclusions: This study demonstrates that using a synchronous fusion strategy with concatenated projections based on attention mechanisms, i.e., speech-to-lips and lips-to-speech, exceeds previous results reported in the literature. Complementary information between lip movement and speech production is confirmed when advanced fusion strategies are employed. Finally, multimodal approaches, combining visual and speech signals, showed great potential to improve PD classification, generating more confident and robust models for clinical diagnostic support.

## 1. Introduction

Parkinson’s disease (PD) is the second most prevalent neurodegenerative disorder worldwide [1]. It is estimated that PD-related treatments exceed USD 4072 only in the USA. Still, this burden can be decreased if the disease progression is better monitored [2]. The research community has been extensively working on developing computational tools for the automatic detection and monitoring of PD. Speech [3] and facial expressions [4] are among the two most commonly used biosignals when creating such technologies. There exist different approaches to evaluating Parkinson’s speech, and similarly when considering facial expressions; however, lip movement has not been extensively explored. Moreover, the combination of these two biosignals has not been considered so far to model symptoms suffered by PD patients. This work focuses on filling this gap by evaluating the suitability of an automatic system that combines speech signals and lip movement to diagnose PD.

**Related works considering speech signals:** There is a large number of works where machine learning (ML)- and deep learning (DL)-based methodologies are used in PD diagnosis and monitoring considering speech signals as the main source of information. Most works are based on representing those signals with a set of Mel Frequency Cepstral Coefficients (MFCCs) [5,6]. Other authors have focused on modeling different speech dimensions, such as articulation, phonation, prosody, and phonemic identifiability [7,8]. Other works propose combining different speech features/dimensions mainly based on classical fusion techniques [9]. Regarding DL-based methods, different architectures have been used to model speech data including Convolutional Neural Networks (CNNs) [10,11] and 1D convolutional layers with Recurrent Neural Networks (RNNs) [12]. Other approaches such as those based on Autoencoders have been used for dimensionality reduction and feature extraction from high-dimensional data, yielding improved classification performance [13]. There are also works based on pre-trained models using transformers trained with large-scale datasets to use afterward, resulting in representations for disease classification [14,15].

**Related works considering speech and lips movement:** There exist works where speech signals are combined with other biosignals for PD classification and monitoring. Typical choices include face, handwriting, gait, and language [14,16]. During the literature revision performed for this paper, we could not find works where speech signals and lip movement information were combined for the automatic discrimination and/or monitoring of PD. To enrich our proposed model, we reviewed related works from other fields where the combination of these two streams was applied. Among the reviewed fields were automatic speech recognition (ASR) and emotion recognition. Results show improvements when lip video streams are combined with uni-modal systems trained with audio signals. According to the authors, such improvements are due to the ability of lip movements to provide visual features that complement the audio ones, especially in cases where speech signals may have noise or are distorted for any reason [17,18,19,20]. There are also works where synchronous fusion techniques based on attention mechanisms are explored. The main idea is to take advantage of their capability to preserve temporal information when fusing both streams while giving greater relevance to discriminative intervals of each modality. These techniques have consistently demonstrated superior performance compared to uni-modal systems [21,22,23,24,25].

For the particular case of PD speech, the research community highlights the presence of motor abnormalities in its production, specifically when pronouncing stop consonants and those phonemes that require controlling the lips, jaw, and tongue [26]. These difficulties observed or perceived in Parkinson’s speech can also be visually observed when having a detailed look at the patient’s mouth movement while speaking [27]. In this paper, we specifically focus on modeling together lip movement and speech production in PD speech. The underlying hypothesis is that the well-known acoustic information encoded in speech signals is effectively complemented with visual/video information extracted from the lip movement.

**Contribution:** We introduce a methodology to discriminate between PD patients vs. HC subjects following an end-to-end architecture based on 1D and 2D CNNs to characterize audio and video signals (lips), respectively. A synchronous combination of both sources of information is performed with appropriately configured attention mechanisms. The resulting bimodal representation is considered for classification between PD patients and HC subjects. The introduced approach is compared with respect to (with respect to) classical approaches that consider hand-crafted features for video and audio signals. These two modalities are later combined using classical fusion methods. The results reported in this paper demonstrate that integrating lip movement information with speech cues improves classification performance, especially when synchronous fusion strategies are employed. These results show the potential of multimodal approaches to capture complementary information that otherwise would not be considered in uni-modal approaches.

## 2. Data

We used the FacePark corpus, which was collected by GITA Lab about 5 years ago. This database originally contained videos of 24 HC subjects and 31 PD patients. However, 1 HC and 1 PD participant were excluded from the analysis because, in most frames, it was not possible to extract the lips from the video. The videos were recorded in non-controlled environment conditions, i.e., light conditions and the background were not controlled prior recording. All the participants were diagnosed according to an expert neurologist and were evaluated according to the Movement Disorder Society-Unified Parkinson’s Disease Rating (MDS-UPDRS-III) scale [28] and the Hoehn and Yahr scale (H&Y) [29]. Each participant was asked to perform different tasks including right eye wink, left eye wink, smile, anger, surprise, and a reading text. In this paper, we only considered the read text because it was the only one that required the participant to speak. Table 1 shows the demographic and clinical information of the participants.

### Data Preprocessing

All recordings were standardized to a frame rate of 7 FPS. Alignment was performed using the roll angle by tracing a straight line between the centers of the eyes to ensure a consistent frontal posture across all video frames. MediaPipe Face mesh, which is an open-source ML platform created by Google, was used to segment the lip region in each video, discarding frames with undetected or incomplete facial regions during preprocessing. For voice signals, each sample was standardized to a sampling rate of 48 kHz. DC offset was removed and amplitude was normalized.

## 3. Methods

Figure 1 illustrates an overview of the proposed methodology. This study implemented a synchronous end-to-end fusion of speech and lip movement features via attention mechanisms. Specifically, speech segments of 1 s of duration and their corresponding lip frames were processed. Feature extraction was performed on each modality while preserving temporal dynamics: a 1D-CNN was used for speech signals, while a ResNet-based 2D-CNN was used to process lip frames. Both encoder networks were trained from scratch and designed with a reduced parameter count to align with the available data limitations. After feature extraction, an attention block was used for fusing one modality over the other while preserving temporal information. We considered both possible scenarios, i.e., speech projected onto the lips and vice versa. Finally, to continue with the classification stage, we performed a temporal pooling to obtain a static representation of the projections. We evaluated three scenarios for classifying PD patients vs. HC subjects: (i) speech-to-lip projection, (ii) lip-to-speech projection, and (iii) concatenated features from both projections. A fully connected layer then consolidated these representations to make the final decision. A baseline was established to compare the proposed methodology with state-of-the-art classical approaches. Comparisons regarding feature extraction and fusion techniques were made. Additional details about each stage are presented below.

### 3.1. Speech Representation

#### 3.1.1. Classical Approach

Different features were computed to model three speech dimensions: articulation, phonation, and prosody, with 488, 28, and 103 features extracted, respectively. The features were extracted using Disvoice, a Python toolbox released at GITA Lab to characterize pathological speech [30]. Articulation features intend to model control over vocal tract movements, particularly during the transition between unvoiced-to-voiced (onset) and voiced-to-unvoiced (offset) segments. This measure reflects the ability of speakers to initiate and stop the movement of the vocal folds [30]. These features include 12 Mel Frequency Cepstral Coefficients (MFCCs), their first and second derivatives, and the energy distribution across the first 17 Bark bands.

Phonation and prosody features further characterize the quality and expressiveness of speech. Phonation features focus on the production of airflow and vocal fold stability. We extracted features based on the variation of the fundamental frequency calculated in short time frames. In addition, jitter, shimmer, and the logarithm of the energy were calculated [31]. Prosody features are based on aspects like fundamental frequency, timing, accent, and intensity. These include the statistical properties of the fundamental frequency (mean, standard deviation, maximum, minimum, skewness, kurtosis) and a 5-degree Lagrange polynomial to model pitch contour [32]. Additionally, several duration features were calculated from the voice signal including voice rate, pause rate, mean, and standard deviation of the duration of the voiced and unvoiced segments, and others [30].

#### 3.1.2. Deep Approach

The deep approach to model speech is based on a 1D-CNN, as 1D convolutional layers are particularly suited for sequential data, enabling the extraction of temporal representations. This layer configuration comprises two main components: the number of channels and the kernel. The kernel functions are filters sliding over the input signal, capturing information based on its learned weights and defined size. Each channel corresponds to a unique kernel, allowing the layer to generate multiple representations from the same input signal channel-wise. The kernel weights are learned during training, while the kernel size is set as a hyper-parameter. The output of a 1D convolutional layer, given an input size of $\left(N,C_{in},L\right)$ and producing an output size of $\left(N,C_{out},L\right)$, can be formally described as
(1)$$\mathrm{out}\left(N_{i},C_{out_{j}},L\right)=\mathrm{bias}\left(C_{out_{j}},L\right)+\sum_{k=0}^{C_{in}-1}\mathrm{weight}\left(C_{out_{j}},k,L\right)★\mathrm{input}\left(N_{i},k,L\right)$$
where ★ is the valid cross-correlation operator, *N* represents the batch size, *C* denotes the number of channels, and *L* is the signal sequence length. In this setup, the kernel functions as a temporal sliding window. To further process the data, temporal max pooling is applied as a down-sampling technique to reduce the temporal resolution of the output. Temporal max pooling operates based on kernel size, determining how many input samples should be combined. As the kernel slides across the data, it computes the maximum value within each segment of samples, producing a condensed output vector. The stride is set to match the kernel length, ensuring non-overlapping pooling windows.

The architecture used in this work was optimized previously for the automatic classification of PD patients vs. HC subjects based on speech signals. It is composed of two convolutional layers with 16–32 channels, and a kernel size of 480–240, respectively. Each layer is followed by a temporal max pooling with a kernel size of 2. The characterization performed by the convolutional layers is the input to a bidirectional LSTM that is responsible for performing the temporal analysis of the network. Finally, the output of the LSTMs feeds a fully connected network with 3 layers to make the final decision. ReLu activations are considered in the convolutional layers, and a Softmax activation function is used at the output. In addition, in order to avoid over-training we considered batch normalization, dropout, L2-regularization, and early stopping techniques. Table 2 details the implemented architecture. Convolutional layers are denoted as (I × O × K, S), where I and O represent the input and output channels, K is the kernel size, and S the stride. The LSTM layer is described as (X × H, N), with X indicating the number of expected input features, H the features in the hidden state, and N the number of recurrent layers. Lastly, fully connected layers are expressed as (X, O), where X is the number of input features and O the output size.

### 3.2. Video Representation

#### 3.2.1. Classical Approach

To characterize the lips area extracted in the preprocessing stage, we used Local Binary Patterns (LBPs). The LBP operator is a texture descriptor introduced by Ojala et al. in [33] to capture local spatial patterns in images. It is grayscale-invariant and robust to changes in contrast. This method operates by comparing each pixel to its neighbors within a circular region. We defined a radius of 1 pixel and considered its 8 pixels around. It is assigned a binary value to each neighbor based on its intensity compared to the corresponding center pixel value. This results in a binary code with the values clockwise concatenated. The code is converted to a decimal value and stored at the center pixel position. When the neighbors are not exactly on the pixel grid positions, bilinear interpolation is used to estimate their intensity values. This circular sampling, together with the flexibility to adjust the radius and number of neighbors, allows the LBP to adapt to different texture scales, making it rotation-invariant and suitable for complex textures. These LBP features are aggregated into histograms that provide compact and highly descriptive representations of texture patterns across the image, forming a feature vector of 255 dimensions to be used for classification.

#### 3.2.2. Deep Approach

We considered a ResNet topology for this study. It is composed of CNNs and it was introduced to address the vanishing gradient problem. The solution utilizes a unique residual learning framework that allows networks to remain accurate and efficient even at greater depths [34].

A standard convolutional layer learns complex mappings from input to output using a non-linear function, often described as y=F(x). However, ResNet introduces a different approach with residual connections, where the output of a block is modified by adding the input directly. This shortcut connection, which creates an identity mapping, forces the network to learn only residual functions, which can be defined as H(x)=F(x)+x, where H(x) represents the original desired mapping, F(x) the residual function learned by the stacked layers, and *x* the identity function. This residual structure has several advantages. First, it does not add extra parameters or computational complexity. Second, given that ResNet layers skip learning direct mappings with F(x)=0, they do not need to approximate the identity function through multiple layers. If the identity mapping is the optimal solution, ResNet adjusts the residuals to zero, making it easier to learn even in deep architectures.

In this work, we designed a ResNet model comprising six residual blocks and three main convolutional stages, featuring 16, 32, and 64 feature maps, respectively. To process the input frames (lips), we first applied a convolutional layer with 16 channels, ensuring compatibility with the first main stage. ReLU activations were incorporated throughout the hidden layers to introduce non-linearity, and a Softmax function in the output layer provided the final classification. After the final block, we performed average pooling across channels to get the representation into 64 features per frame, which were then processed by a fully connected layer to classify each frame as PD or HC. As in the speech representation based on a deep approach, we considered batch normalization, dropout, L2-regularization, and early stopping techniques. In addition, Table 3 shows details of the architecture used for lip modeling.

### 3.3. Fusion Strategies

Several fusion methods were implemented in this work. The first strategy comprises three classical methods: early, late, and joint. The second strategy is based on attention blocks that preserve temporal dominance during the speech-to-lip fusion. Details of each fusion are presented below.

#### 3.3.1. Classical Fusion Methods

*Early fusion:* This is a traditional approach where data from multiple sources are combined at the beginning of the analysis process. This method is applied to raw or preprocessed data, often requiring the extraction of features beforehand. This is very convenient, especially when modalities have different sampling rates. The main drawback of early fusion is the potential loss of information because each modality needs to be represented as a static feature vector. Additionally, synchronizing timestamps across modalities can be challenging.

*Late fusion:* In contrast to the previous method, it assumes independence among all sources of information and performs the fusion itself at the decision-making stage. This method is robust against diverse sampling frequencies among sources of information and dimensionality. This approach could offer better performance because the errors from various models are processed independently. Among the most commonly used methods for combining the results of independent models are maximum fusion, mean fusion, and Bayes rules.

*Joint fusion:* Also known as intermediate fusion, it takes advantage of deep neural network architectures by combining features extracted from intermediate layers of different modalities [35]. Unlike early and late fusions, this method incorporates back-propagation, meaning that the loss is propagated back to the feature extraction layers, allowing the network to refine feature representations using information from all modalities. Joint fusion can be performed by fusing multiple modalities into a single shared representation layer or progressively incorporating one or more modalities. Notice that merging features from different modalities into a single layer can increase representational richness; however, it also increases the risk of overfitting and can hinder the network’s ability to capture accurately cross-modal relationships.

#### 3.3.2. Synchronous Fusion

To combine flow information without losing temporal dominance, we considered attention mechanisms. Although it is not a standard fusion modality, state-of-the-art works have used it as a fusion method. The attention mechanism was introduced by Bahdanau in [36] to address the bottleneck problem that emerges with the use of fixed-length encoding vectors, where the decoder would have limited access to the information provided by the input. Attention mechanisms allow the decoder to use the most relevant parts of the input sequence through a weighted combination of all the encoded input vectors. The most relevant ones are attributed the highest weights. Attention layers have been empirically demonstrated to be effective in modeling language sequences and have become indispensable [37].

Attention layers are fundamentally a weighted mean reduction. Attention is unusual among layers because it takes three inputs, whereas most layers in deep learning take just one or perhaps two. These inputs are called queries (Q), values (V), and keys (K) (see Equation (Equation 2)). The reduction occurs over the values; therefore, if the values are ranked 3, the output will be ranked 2. The query should be one below the keys. The keys should be of the same rank as the values. The keys and query determine how to weigh the values according to the attention mechanism.
(2)$$Attention\left(Q,K,V\right)=softmax\left(\frac{QK^{T}}{\sqrt{d_{k}}}\right)V$$

The dot product between Q and $K^{T}$ has information about the relationships among elements of Q and elements of K. Then, this matrix is scaled by the $\sqrt{d_{k}}$ and a softmax function is applied to obtain weights, known as attention scores. Finally, the output is computed as a weighted sum of the values V, where the weight assigned to each value is computed by a compatibility function of the query with the corresponding key. For example, in [38], a transformer-based audiovisual fusion was performed; for this case, K and V were assigned to the features obtained from an encoder using the videos as input, while Q was the characterization of the speech signal. K and V may be considered keys and values in a “soft” dictionary, while Q is a query that contextualizes the attention weights.

After obtaining each representation, we used the attention mechanism to synchronously combine the speech stream with the lip stream considering the temporal domain (see Figure 2). After obtaining this new representation containing both streams, we computed an average in the temporal dimension to obtain a static representation and proceed with the classification stage. In this work, we performed the speech-to-lip and lip-to-speech projections, and also both, after temporal pooling (speech-to-lip + lip-to-speech).

## 4. Results

The experimental setup included classical and deep approaches to evaluate the performance of different fusion techniques. For classical analysis, individual models were implemented for speech and lip data: a 1D-CNN was applied to the speech modality, while a ResNet with three main blocks was used to process the lip frames. In addition, classical fusion techniques, including early, late, and joint fusions, were employed to combine data from multiple modalities. For the deep approach, a synchronous fusion was explored through various projection strategies: speech-to-lip, lip-to-speech, and their combination, namely speech-to-lip and lip-to-speech projection. Each experiment was trained and evaluated following the same 5-fold speaker-independent stratified cross-validation strategy. The results are reported in terms of mean and standard deviation computed along the folds.

### 4.1. Classical Approach

Different uni-modal approaches were evaluated. The LBP was used to characterize the lips and different dimensions were used to model the speech signals: prosody, phonation, articulation, and their fusion. Then, we considered the best representations and combined both approaches using early and late fusion. In all the experiments (except for the late-mean), we used a Support Vector Machine (SVM) for the classification of PD vs. HC. In the case of the late-mean, we calculated the mean of the scores and the prediction was performed based on a threshold.

Table 4 shows the results of the classical uni-modal approaches, and also for the early and late fusions. We can observe that in the uni-modal approaches, the combination of the three speech dimensions yielded the best result with 86.2% of UAR. This allows us to conclude that they are complementary dimensions, as reported in previous work on the classification of PD. When we evaluated the complementarity of the lips information projected on speech, the best result was obtained with the late-mean technique with a UAR of 86.8%, obtaining a similar performance as with the speech stream alone, but with a higher imbalance between sensitivity and specificity. Therefore, we could conclude that for classical representations, lip characterization does not generate complementary information suitable for PD classification.

### 4.2. Deep Approach

In the deep approach, we initially trained and evaluated uni-modal models for the two streams: 2D-CNNs to model lip frames and 1D-CNNs to model speech. In both cases, we performed a 1 s alignment between both streams. Furthermore, since each participant pronounced several utterances, the final decision was made by computing the mode along the predictions. Therefore, we obtained a static representation of each stream to evaluate different classical fusion techniques without dealing with the time domain. Finally, we evaluated the synchronous fusion based on attention mechanisms between the two streams. For this, we evaluated the projection of speech on the lips, lips on speech, and the concatenation of both projections. The results obtained are reported in Table 5.

For the uni-modal approaches, the speech-based model achieved the highest UAR of 90%, while the lip-based model reached 88%. As shown in Table 5, the classical fusion methods yielded results comparable to those obtained with the uni-modal speech model; however, they had a reduced standard deviation in certain metrics. Overall, these results indicate that classical fusion methods do not improve the performance of uni-modal approaches.

When we analyzed the synchronous fusion, we can observe that projecting the speech stream onto the lip stream yielded results similar to the uni-modal lip model, indicating limited effectiveness of the attention mechanism in this direction. However, when we projected the lip stream on the speech stream, we observed a UAR improvement of around 3% compared to the uni-modal speech approach. This allows us to conclude that the lips provide information complementary to that extracted from the speech signals.

Finally, when both projections are concatenated, the UAR increases to 95%, with a remarkable reduction in the standard deviation of certain metrics. These results suggest that the lip stream, which shows lower performance in uni-modal evaluations, improves the discrimination capability of the speech modality, but not vice versa. This is probably due to their ability to capture visual cues associated with articulation and mouth movements. These visual cues, although less effective independently, can capture variations in mouth control and precision that reinforce the speech model’s ability to distinguish patterns in Parkinson’s speech.

We are aware of results that are in line with our study but reported in areas of automatic speech recognition and emotion recognition [18,22,23]; however, to the best of our knowledge, this approach has not been evaluated in Parkinson’s speech classification. Our findings show the potential of integrating visual information from lip movements as a new research avenue for improving pattern recognition models applied to medical diagnosis. Further exploration of this approach could pave the way for the development of robust models suitable in clinical settings.

## 5. Discussion

Research on PD has begun recently to incorporate multimodal approaches to improve performance and robustness. However, studies combining video and speech signals remain limited mainly due to data scarcity. According to our research, only two papers addressed this fusion for PD classification. In [39], facial geometric features and speech attributes related to volume and pitch variations were combined to classify 74 PD patients versus 74 HC subjects, achieving an AUC of 0.85. Similarly, [40] proposed a joint analysis of hypomimia and hypokinetic dysarthria to diagnose PD. The study considered 73 PD patients and 46 HC participants and the authors reported accuracy improvements from 77% with speech and 81% with video, to 83% with the fusion.

Direct comparisons between those studies in the state-of-the-art and our work is not possible due to differences in the corpora. Notice that the two studies mentioned above employed classical fusion techniques, i.e., they primarily integrated static features or decisions without taking advantage of temporal dynamics in multimodal interactions. In contrast, our study introduces a novel synchronous fusion methodology based on attention mechanisms to integrate speech and lip movement features. This approach preserves temporal information and identifies and prioritizes discriminative intervals across both modalities. Our results demonstrate that this strategy outperforms classical fusion methods and uni-modal approaches, achieving a UAR of 95%.

Our results show the potential of attention-based synchronous fusion for assessing the complementarity of audiovisual streams. Furthermore, this work opens new avenues of research on synchronous fusion for evaluating and monitoring PD, showing the way for more effective and dynamic clinical applications. For instance, studies about which specific areas of the face, including specific articulators and limbs [41,42], correlate the most with synchronous speech production will solve open research questions and generate new interesting hypotheses regarding speech production and face movement in people suffering from movement disorders.

## 6. Limitations and Future Work

This study presents several limitations that must be addressed in future research to enhance the robustness and applicability of the proposed methodologies. The first limitation lies in the temporal alignment strategy. The use of 1-s segments may restrict the model’s ability to capture long-term temporal dependencies which may be important in multimodal interactions. The exploration of alternative alignment approaches, such as dynamic segmentation, could improve the system’s performance and robustness [43]. Another limitation is the fact that this work focuses exclusively on characterizing lip movements, overlooking the potential contribution of other facial features/areas such as eyes, eyebrows, or cheeks, which may provide complementary information for detecting PD. In future work, we will expand the range of facial analysis to incorporate representations of the complete face to improve the model’s performance by capturing richer visual features. Finally, integrating interpretable techniques such as attention heat maps, would provide valuable information on the model’s decision-making process. Although in the development of this work we analyzed these heat maps, we did not achieve conclusive results. We believe that increasing the data size, accompanied by other representations, will enable interpretable results.

## 7. Conclusions

This study proposed a novel methodology for the synchronous fusion of speech and lip movement features to classify PD patients and HC subjects. The approach integrates deep learning-based 1D-CNN and 2D-CNN models to extract features from audio and video modalities, respectively, and employs attention mechanisms to perform temporal fusion. The best UAR of 95.8% was achieved when using a synchronous fusion strategy with concatenated projections, i.e., speech-to-lips and lips-to-speech. This performance exceeds classical baselines and classical fusion techniques in the deep learning framework, highlighting the effectiveness of the proposed attention-based fusion. This study demonstrates that lip movement information provides complementary information to speech signals, especially when advanced fusion strategies are employed. This confirms that multimodal approaches, combining visual and speech signals, have great potential to improve PD classification, generating more confident and robust models for clinical diagnostic support.

## Figures and Tables

**Figure 1 diagnostics-15-00073-f001:**
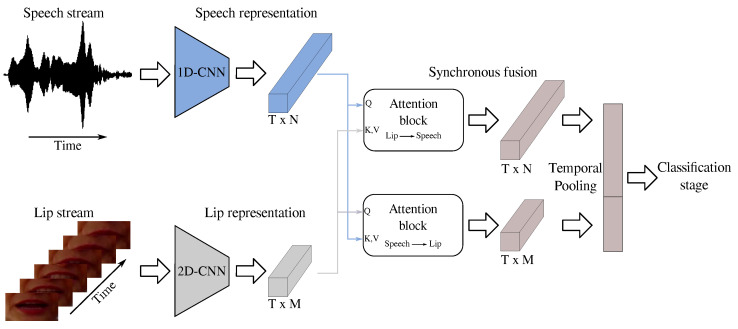
General methodology proposed in this study.

**Figure 2 diagnostics-15-00073-f002:**
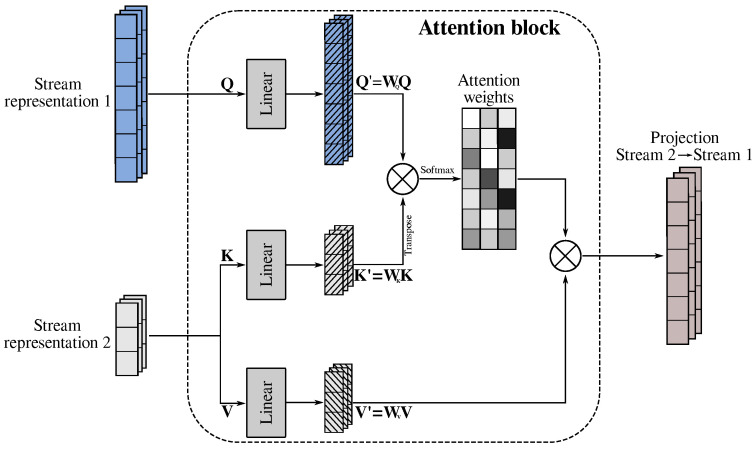
Structure of the attention block.

**Table 1 diagnostics-15-00073-t001:** Demographic and clinical information of the participants included in the FacePark-GITA database. Values are reported in terms of mean ± standard deviation.

	PD Patients		HC Subjects		PD vs. HC
	Male	Female	Male	Female	
Number of participants	18	12	11	12	* *p* = 0.54
Age	69.1 ± 9.5	67.4 ± 10.9	63.9 ± 9.5	65.3 ± 8.7	** *p* = 0.15
Range of age	52–86	53–87	49–79	49–83	
Time since diagnosis	8.7 ± 5.5	15.6 ± 17.3			
MDS-UPDRS-III	34.4 ± 13.6	29.7 ± 12.3			
Hoehn and Yahr scale	2.3 ± 0.4	2.5 ± 0.5			

* *p*-value calculated through Chi–square test. ** *p*-value calculated through *t*-test.

**Table 2 diagnostics-15-00073-t002:** Architecture used for speech modeling. Conv.: convolution; MaxPool.: max pooling; FC: fully connected.

Stage	Type of Layer	Output Size
Conv. 1	Conv. (1 × 16 × 480,12)	3961 × 16
Pooling 1	MaxPool. (2)	1980 × 16
Conv. 2	Conv. (16 × 32 × 240,6)	291 × 32
Pooling 2	MaxPool. (2)	145 × 32
LSTM	LSTM (32 × 64,2)	145 × 64
FC1	FC (9280,1024)	1024
FC2	FC (1024,256)	256
Output	FC (256,2)	2

**Table 3 diagnostics-15-00073-t003:** Architecture used for lip modeling. Conv.: convolution; Avg. Pool.: average pooling; FC: fully connected.

Stage	Type of Layer		Output Size
Conv. 1	Conv. (1 × 16 × 3,1)		100 × 50 × 16
Block 1	Conv. (16 × 16 × 3,1)	} × 2	100 × 50 × 16
	Conv. (16 × 16 × 3,1)		
Block 2	Conv. (16 × 32 × 3,2)	} × 2	50 × 25 × 32
	Conv. (32 × 32 × 3,1)		
Block 2	Conv. (32 × 64 × 3,2)	} × 2	25 × 13 × 64
	Conv. (64 × 64 × 3,1)		
Pooling	Avg. Pool. (25,13)		64
Output	FC (64,2)		2

**Table 4 diagnostics-15-00073-t004:** Classification between PD patients and HC subjects from speech and lip streams using uni-modal approaches and classical fusion techniques.UAR: unweighted average recall; Sens.: sensitivity; Spec.: specificity. Results reported in terms of mean ± standard deviation.

Stream	Technique	UAR (%)	Sens. (%)	Spec. (%)	F1score (%)
Uni-modal approach					
Lip	LBP	78.2 ± 11.5	83.3 ± 14.9	73.0 ± 11.7	79.0 ± 11.5
Speech	Prosody	64.5 ± 9.3	90.0 ± 13.3	39.0 ± 15.0	64.5 ± 9.5
	Phonation	62.0 ± 14.6	80.0 ± 12.5	44.0 ± 20.8	62.5 ± 14.8
	Articulation	82.8 ± 12.7	86.7 ± 12.5	79.0 ± 22.0	82.5 ± 12.8
	Full	86.2 ± 3.9	93.3 ± 13.3	79.0 ± 12.8	86.4 ± 4.9
Fusion					
Lip + Speech	Early	85.2 ± 11.0	93.3 ± 8.2	77.0 ± 16.0	86.3 ± 10.4
	Late	84.8 ± 9.1	86.7 ± 26.7	83.0 ± 8.7	83.7 ± 13.5
	Late-mean	86.8 ± 11.0	96.7 ± 6.7	77.0 ± 16.0	88.0 ± 10.2

**Table 5 diagnostics-15-00073-t005:** Classification between PD patients and HC subjects from speech and lip streams using deep uni-modal approaches, classical fusion techniques, and synchronous fusion based on an attention mechanism. UAR: unweighted average recall; Sens.: sensitivity; Spec.: specificity. Results reported in terms of mean ± standard deviation.

Stream	Technique	UAR (%)	Sens. (%)	Spec. (%)	F1score (%)
Uni-modal approach					
Lip	2D CNN	88.5 ± 12.1	90.0 ± 14.9	87.0 ± 12.0	88.7 ± 12.2
Speech	1D CNN	90.5 ± 8.9	90.0 ± 14.9	91.0 ± 12.5	90.3 ± 9.6
Classical Fusion					
Lip + Speech	Early	90.5 ± 5.5	90.0 ± 13.3	91.0 ± 11.1	90.3 ± 6.3
	Joint	86.3 ± 12.5	86.7 ± 13.9	86.0 ± 21.9	86.3 ± 11.7
	Late	89.7 ± 13.5	93.3 ± 8.2	86.0 ± 19.6	90.2 ± 12.7
	Late-mean	90.2 ± 9.6	91.3 ± 8.2	89.0 ± 11.1	90.4 ± 9.4
Synchronous fusion					
Lip + Speech	Speech-to-lip	88.3 ± 11.7	88.7 ± 12.5	88.0 ± 11.6	88.4 ± 11.8
	Lip-to-speech	93.3 ± 7.8	88.7 ± 12.5	98.0 ± 8.0	92.4 ± 8.6
	Both	95.8 ± 5.3	96.7 ± 6.7	95.0 ± 10.0	96.0 ± 4.9

## Data Availability

The data considered in this work are not publicly available.

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
