# Peer review of "Synchronous Analysis of Speech Production and Lips Movement to Detect Parkinson’s Disease Using Deep Learning Methods"

_diagnostics, 2024, doi:10.3390/diagnostics15010073_

Round 1
Reviewer 1 Report
Comments and Suggestions for Authors
A methodology using speech and video images to distinguish Parkinson's patients from healthy individuals is proposed. The dataset and the proposed methodology are detailed. Three different scenarios have been implemented and performance parameters have been calculated. I recommend the following corrections to be considered.
1. The problem has two classes. These are Parkinson's disease and HC. HC is not defined in the article. Does HC represent healthy individuals?
2. Please add the layered structure and parameters of the 1D-CNN and 2D-CNN used to the article with a table.
3. It is stated in the limitation section that the dataset size is not sufficient. This is the biggest limitation of the article. I think there is not enough data for training, testing, and validation. I think the learning process is not generalized. There are 53 data in total. If 30% is used for testing, 16 data are used. Misclassifying any of these data has a 6.25% effect on accuracy. This situation will be clearer when looking at the confusion matrix. The dataset should be augmented to eliminate this disadvantage. If any patient's speech and video images are divided into 2 parts, you will have a total of 106 data. Data augmentation can be added to the article or this disadvantage can be discussed.
4. A discussion section can be added to compare the obtained results with the literature.
5. It is appropriate to give details about feature fusion. How many features were used? Which classification algorithm was used?
6. If you have recorded to understand the training process, add the loss function change for the scenarios.
7. In line 307 the table number is uncertain.
Author Response
Medellín
24th of December, 2024
Dear Editor and Reviewers,
We want to thank you for your support in the review process of the manuscript diagnostics-3384095 entitled “Synchronous Analysis of Speech Production and Lips Movement to Detect Parkinson’s Disease Using Deep Learning Methods”.
We also thank the reviewers for their time, constructive comments, support and dedication in reviewing the manuscript. We acknowledge that their comments helped to improve the manuscript. Please, find below the answers to the reviewer’s comments. All corrections were considered and highlighted in blue in the corrected manuscript.
Reviewer 1
Comments and Suggestions for Authors
A methodology using speech and video images to distinguish Parkinson's patients from healthy individuals is proposed. The dataset and the proposed methodology are detailed. Three different scenarios have been implemented and performance parameters have been calculated. I recommend the following corrections to be considered.
1. The problem has two classes. These are Parkinson's disease and HC. HC is not defined in the article. Does HC represent healthy individuals?
Answer: The reviewer is correct, HC represents healthy controls. The acronym has been already defined in the manuscript. Thanks for this correction.
2. Please add the layered structure and parameters of the 1D-CNN and 2D-CNN used to the article with a table.
Answer: All details about layered structure and parameters have been added to the corrected manuscript in two new tables in Section 3.1.2 and Section 3.2.2.
3. It is stated in the limitation section that the dataset size is not sufficient. This is the biggest limitation of the article. I think there is not enough data for training, testing, and validation. I think the learning process is not generalized. There are 53 data in total. If 30% is used for testing, 16 data are used. Misclassifying any of these data has a 6.25% effect on accuracy. This situation will be clearer when looking at the confusion matrix. The dataset should be augmented to eliminate this disadvantage. If any patient's speech and video images are divided into 2 parts, you will have a total of 106 data. Data augmentation can be added to the article or this disadvantage can be discussed.
Answer: Although the sample size is not as big as in other topics like speech recognition or speaker identification, we are cautious about minimizing potential sources of bias. We are aware that a model with thousands of participants would be more robust, unfortunately, that sample size is not realistic/feasible with the current technological conditions around the world. The sample size used in this paper is competitive compared to other studies in the state of the art, additionally, all our models were trained and optimized following a 5-fold speaker-independent stratified cross-validation, which effectively minimizes the risk of overfitting.
Regarding the generalization capability of our models, we used 1s windows to guarantee a dynamic approach along the experiments. Therefore, we have around 30 samples per speaker, for a total of 1060 samples in the experimental setup, resulting in subject-wise models more robust.
We are not aware of other studies in the state of the art where video and speech are synchronously combined. Therefore, even though the sample-size limitation, this paper constitutes a real contribution and a step forward in modeling both bio-signals collected from PD patients. There are a few studies in the state of the art where these two bio-signals are combined non-synchronously, but synchronous studies were not found in our review. The importance of making the analyses synchronously relies on the fact that it enables the study of correlations between specific muscles and limbs that are moved in the vocal tract and in the external part of the lips (and the face) while producing speech. Note that we are aware that, in future studies, it will be necessary to extend the experiments and consider other areas of the face. In this paper we wanted to follow a well-known approach in speech recognition, where the specific area of the lips is considered when making the fusion between acoustics and video.
4. A discussion section can be added to compare the obtained results with the literature.
Answer: Thanks for this suggestion. We acknowledge that a discussion section improves the readability of the manuscript and helps readers to make themselves a complete picture of the topic. Therefore, the discussion has been added in Section 5 of the corrected manuscript.
5. It is appropriate to give details about feature fusion. How many features were used? Which classification algorithm was used?
Answer: The number of features resulting in each approach is now indicated in the corrected manuscript. Section 3.1.1. mentions the dimensionality of the feature vectors for speech representation. Section 3.2.1. mentions dimensionality for the case of video representations. Regarding the classifiers, we used an SVM for the classical approaches and a fully connected network for the deep approach. The use of the fully connected network is indicated in the methodology, where we wanted to make more emphasis on the contribution; the use of the SVM is mentioned in Section 4.1.
6. If you have recorded to understand the training process, add the loss function change for the scenarios.
Answer: Unfortunately, we did not record that data during the training process. We made sure of minimizing overfitting by incorporating several strategies including dropout, L2 regularization, batch normalization, and early stopping. This is clarified (mentioned) in Section 3.2.2 of the corrected manuscript. Thanks for pointing out this relevant detail.
7. In line 307 the table number is uncertain.
Answer: There was a typo. Thanks for pointing this out.
NOTE: See the corrected manuscript in the attachment.
Juan Rafael Orozco Arroyave, Msc, PhD
Full Professor
Head of GITA Lab
Faculty of Engineering
University of Antioquia, Medellín, Colombia

Reviewer 2 Report
Comments and Suggestions for Authors
These authors have produced a novel machine learning approach that performed fusion of information extracted from recordings of subjects reading aloud and corresponding videos of lip movement to classify Parkinson’s disease (PD) patients and healthy control (HC) subjects. Convolutional neural network (1D-CNN and 2D-CNN) models were trained to extract features from audio and video modalities, respectively, and attention-based temporal fusion was performed. The best unweighted average recall (UAR) of 95.8% was achieved by using a synchronous fusion strategy with a concatenated projection of speech-to-lips. This performance exceeded those of classical baselines and fusion techniques with deep learning frameworks. It is concluded that lip movement information provides complementary information to speech signals when classifying subjects as PD and HC. The bimodal ML model is more accurate and suitable than uni-modal approaches or classical asynchronous approaches that combine both sources of information but do not incorporate the underlying temporal information.
This is a novel and interesting paper but the algorithms were trained using only 30 clinically diagnosed PD patients with well established disorders and 23 HCs. These PD patients would not be problematic to diagnose – what is needed is a validation of the authors’ ML approach when used to classify early grey cases as PD or HC. The PD cases classified here were clinically diagnosed without imaging support or seed amplification studies so some may have had atypical syndromes. Interestingly, these workers have concentrated on lip movement rather than extracting all facial feature such as slow blinking and poverty of facial expression. Having said that, the authors’ bimodal fusion approach is clearly a step forward and will no doubt be developed further.
Author Response
Medellín
24th of December, 2024
Dear Editor and Reviewers,
We want to thank you for your support in the review process of the manuscript diagnostics-3384095 entitled “Synchronous Analysis of Speech Production and Lips Movement to Detect Parkinson’s Disease Using Deep Learning Methods”.
We also thank the reviewers for their time, constructive comments, support and dedication in reviewing the manuscript. We acknowledge that their comments helped to improve the manuscript. Please, find below the answers to the reviewer’s comments. All corrections were considered and highlighted in blue in the corrected manuscript.
Reviewer 2
Comments and Suggestions for Authors
These authors have produced a novel machine learning approach that performed fusion of information extracted from recordings of subjects reading aloud and corresponding videos of lip movement to classify Parkinson’s disease (PD) patients and healthy control (HC) subjects. Convolutional neural network (1D-CNN and 2D-CNN) models were trained to extract features from audio and video modalities, respectively, and attention-based temporal fusion was performed. The best unweighted average recall (UAR) of 95.8% was achieved by using a synchronous fusion strategy with a concatenated projection of speech-to-lips. This performance exceeded those of classical baselines and fusion techniques with deep learning frameworks. It is concluded that lip movement information provides complementary information to speech signals when classifying subjects as PD and HC. The bimodal ML model is more accurate and suitable than uni-modal approaches or classical asynchronous approaches that combine both sources of information but do not incorporate the underlying temporal information.
This is a novel and interesting paper but the algorithms were trained using only 30 clinically diagnosed PD patients with well established disorders and 23 HCs. These PD patients would not be problematic to diagnose – what is needed is a validation of the authors’ ML approach when used to classify early grey cases as PD or HC. The PD cases classified here were clinically diagnosed without imaging support or seed amplification studies so some may have had atypical syndromes. Interestingly, these workers have concentrated on lip movement rather than extracting all facial feature such as slow blinking and poverty of facial expression. Having said that, the authors’ bimodal fusion approach is clearly a step forward and will no doubt be developed further.
Answer: Thanks for this positive comment about our work. For the sake of clarification, we focused on modeling the limbs and muscles that interact the most during speech production, thats why we decided to focus on lip movement rather than the entire face. The main motivation came from previous studies on speech recognition with video and speech signals (References 21 to 25 in the corrected manuscript).

Reviewer 3 Report
Comments and Suggestions for Authors
This study introduces a compelling approach to Parkinson’s disease (PD) detection by integrating speech production and lip movement data through deep learning methods. The results demonstrate that lip movement provides valuable complementary information to speech data, significantly enhancing classification accuracy. Notably, the synchronous fusion method achieves an impressive unweighted average recall (UAR) of 95.8%, outperforming classical fusion approaches and showcasing the potential of multimodal integration.
While the study highlights the promise of multimodal approaches for PD detection and monitoring, I believe the authors should elaborate on the rationale for prioritizing lip movements over other facial dynamics or articulatory features, such as tongue or jaw movements. This could involve a brief discussion of how PD impacts specific articulatory and motor control deficits at different stages of the disease. For instance, Longeman (1978) found that speech and voice symptoms in 200 Parkinson’s patients could be grouped into five categories, with 45% showing only laryngeal dysfunction and others exhibiting progressively more complex issues involving the back tongue, tongue blade, lips, and tongue tip. In addition, Wayland et al. (2023) found that muscles controlling tongue tip movements might be more affected than those governing lip movement in individuals with PD.
Furthermore, I think the authors should demonstrating the model’s classification performance in relation to disease severity, using clinical scales such as the MDS-UPDRS-III or Hoehn and Yahr scale, would enhance the clinical relevance of this work. Such an analysis could provide insights into the model’s ability to detect PD across various stages of progression, which is critical for its potential application in real-world settings.
Minor comment:
Page 4, lines 13–133: The phrase "These include the statistical properties of the fundamental frequency (mean, standard deviation, maximum, minimum, skewness, kurtosis) and a 5-degree Lagrange polynomial to model the pitch contour??" includes a '??' that appears to be a placeholder or error. Please clarify its meaning or revise for clarity.
Reference
1. Logemann, J. A., Fisher, H. B., Boshes, B., & Blonsky, E. R. (1978). Frequency and cooccurrence of vocal tract dysfunctions in the speech of a large sample of Parkinson patients. Journal of Speech and hearing Disorders, 43(1), 47-57.
2. Wayland, R., Tang, K., Wang, F., Vellozzi, S., Meyer, R., & Sengupta, R. (2023, December). Neural network-based measure of consonant lenition in Parkinson's Disease. Proceedings of Meetings on Acoustics, 52(1). AIP Publishing.
Author Response
Medellín
24th of December, 2024
Dear Editor and Reviewers,
We want to thank you for your support in the review process of the manuscript diagnostics-3384095 entitled “Synchronous Analysis of Speech Production and Lips Movement to Detect Parkinson’s Disease Using Deep Learning Methods”.
We also thank the reviewers for their time, constructive comments, support and dedication in reviewing the manuscript. We acknowledge that their comments helped to improve the manuscript. Please, find below the answers to the reviewer’s comments. All corrections were considered and highlighted in blue in the corrected manuscript.
Reviewer 3
Comments and Suggestions for Authors
This study introduces a compelling approach to Parkinson’s disease (PD) detection by integrating speech production and lip movement data through deep learning methods. The results demonstrate that lip movement provides valuable complementary information to speech data, significantly enhancing classification accuracy. Notably, the synchronous fusion method achieves an impressive unweighted average recall (UAR) of 95.8%, outperforming classical fusion approaches and showcasing the potential of multimodal integration.
While the study highlights the promise of multimodal approaches for PD detection and monitoring, I believe the authors should elaborate on the rationale for prioritizing lip movements over other facial dynamics or articulatory features, such as tongue or jaw movements. This could involve a brief discussion of how PD impacts specific articulatory and motor control deficits at different stages of the disease. For instance, Longemann (1978) found that speech and voice symptoms in 200 Parkinson’s patients could be grouped into five categories, with 45% showing only laryngeal dysfunction and others exhibiting progressively more complex issues involving the back tongue, tongue blade, lips, and tongue tip. In addition, Wayland et al. (2023) found that muscles controlling tongue tip movements might be more affected than those governing lip movement in individuals with PD.
Answer: Thanks for this positive comment about our work. For the sake of clarification, we focused on modeling the limbs and muscles that interact the most during speech production, that’s why we decided to focus on lip movement rather than the entire face. The main motivation came from previous studies on speech recognition with video and speech signals (References 21 to 25 in the corrected manuscript).
Thanks to the suggestion of the reviewer, we have added two more references to the manuscript (Longeman et al. 1978 and Wayland et al. 2023) to enrich the discussion about whether to focus on the lip area or to include the whole face. This discussion has been added to the Discussion section.
Furthermore, I think the authors should demonstrating the model’s classification performance in relation to disease severity, using clinical scales such as the MDS-UPDRS-III or Hoehn and Yahr scale, would enhance the clinical relevance of this work. Such an analysis could provide insights into the model’s ability to detect PD across various stages of progression, which is critical for its potential application in real-world settings.
Answer: Sample-wise makes this experiment not feasible. We plan to record more subjects during 2025 (Q3).
Minor comment:
Page 4, lines 13–133: The phrase "These include the statistical properties of the fundamental frequency (mean, standard deviation, maximum, minimum, skewness, kurtosis) and a 5-degree Lagrange polynomial to model the pitch contour??" includes a '??' that appears to be a placeholder or error. Please clarify its meaning or revise for clarity.
Answer: Thanks for pointing this out. There was a typo and we already corrected it.
NOTE: See the corrected manuscript in the attachment.
Juan Rafael Orozco Arroyave, Msc, PhD
Full Professor
Head of GITA Lab
Faculty of Engineering
University of Antioquia, Medellín, Colombia

Round 2
Reviewer 1 Report
Comments and Suggestions for Authors
Thank you for your answers.